# Plasmon-enhanced stimulated Raman scattering microscopy with single-molecule detection sensitivity

Cheng Zong[1,2], Ranjith Premasiri[3,4], Haonan Lin[1], Yimin Huang[3], Chi Zhang[1], Chen Yang[3,4], Bin Ren [2], Lawrence D. Ziegler[3,4] & Ji-Xin Cheng[1,3,4]*

Stimulated Raman scattering (SRS) microscopy allows for high-speed label-free chemical imaging of biomedical systems. The imaging sensitivity of SRS microscopy is limited to ~10 mM for endogenous biomolecules. Electronic pre-resonant SRS allows detection of sub-micromolar chromophores. However, label-free SRS detection of single biomolecules having extremely small Raman cross-sections (~$10^{-30}$ cm$^2$ sr$^{-1}$) remains unreachable. Here, we demonstrate plasmon-enhanced stimulated Raman scattering (PESRS) microscopy with single-molecule detection sensitivity. Incorporating pico-Joule laser excitation, background subtraction, and a denoising algorithm, we obtain robust single-pixel SRS spectra exhibiting single-molecule events, verified by using two isotopologues of adenine and further confirmed by digital blinking and bleaching in the temporal domain. To demonstrate the capability of PESRS for biological applications, we utilize PESRS to map adenine released from bacteria due to starvation stress. PESRS microscopy holds the promise for ultrasensitive detection and rapid mapping of molecular events in chemical and biomedical systems.

[1] Department of Electrical and Computer Engineering, Department of Biomedical Engineering, Boston University, Boston, MA 02215, USA. [2] State Key Laboratory of Physical Chemistry of Solid Surfaces, MOE Key Laboratory of Spectrochemical Analysis and Instrumentation, Collaborative Innovation Center of Chemistry for Energy Materials, College of Chemistry and Chemical Engineering, Xiamen University, 361005 Xiamen, China. [3] Department of Chemistry, Boston University, Boston, MA 02215, USA. [4] Photonics Center, Boston University, Boston, MA 02215, USA. *email: jxcheng@bu.edu

Raman spectroscopy is a versatile analytical tool providing information about the native fingerprint vibrational states of a sample determined by its molecular structure and chemical environment. Non-electronically resonant spontaneous vibrational Raman scattering cross-sections are typically $10^{-30}$ cm$^2$ sr$^{-1}$ and intrinsically small cross-sections on this order result in detection limits only as low as milli-molar (mM) levels. By placing a molecule close to a plasmonic nanostructure, plasmon-enhanced Raman spectroscopy pushes the detection sensitivity to the single-molecule level[1–9], yet the speed in spectral acquisition is still not sufficient for ultrasensitive chemical mapping of molecular events in a dynamic and complex system[10].

Owing to the development of advanced lasers and electro-optic instruments, nonlinear Raman microscopy has been shown to provide label-free chemical imaging, based on either coherent anti-Stokes Raman scattering (CARS) or stimulated Raman scattering (SRS), for a broad range of biomedical applications[11]. Early developments of CARS and SRS microscopy relied on picosecond pulses for detection of a single Raman peak[12–17]. Intra-pulse broadband CARS, developed by Cicerone and coworkers, allowed recording of a whole Raman spectrum within 3.5 ms[18]. Hyperspectral SRS microscopy has been achieved recently by many strategies, such as wavelength tuning[19,20], spectral-focusing[21,22], optical frequencies coding[23], etc, which provide spectral profile at each image pixel and enable the discoveries of new biology[24,25]. Multiplex SRS microscopy developed by Cheng and coworkers[26], is able to acquire a Raman spectrum covering a 200 wavenumber spectral window within 5 µs, which allowed high-throughput chemical analysis in a flow cytometry setting[27]. Yet, the imaging sensitivity of SRS microscopy is limited to ~10 mM for chemical bonds such as the C–H vibrations in cell membranes[22,28]. Min and coworkers recently reported electronic pre-resonance SRS achieving sub-micromolar-sensitivity detection for chromophores having a Raman cross-section $10^3$ or $10^4$ times larger than endogenous biomolecules[29,30].

To push coherent Raman detection sensitivity further, plasmon-enhanced CARS has been reported[31–37] and single-molecule sensitivity has been proved[32,34,35]. While, the CARS signal displays a nonlinear dependence on the concentration of analytes[38]. The SRS signal, on the other hand, shows a linear dependence on the concentration of analytes[17]. The Van Duyne group reported reproducible surface-enhanced femtosecond SRS spectra from molecules embedded in a gold nano-dumbbell sol[39,40]. Yet, plasmon-enhanced SRS at single-molecule detection sensitivity has not been reported. Major hurtles of achieving single-molecule SRS detection include the damage of plasmonic substrates by the ultrafast pulses[41] and a large pump-probe background, arising from plasmon-induced photothermal and/or stimulated emission process.

Here, we report plasmon-enhanced SRS (PESRS) microscopy (Fig. 1a, instrument in Supplementary Fig. 1) and its application to ultrasensitive imaging of biomolecules released from cells. We reach single-molecule detection sensitivity by incorporating several innovations. First, we use chirped laser pulses at 80 MHz repetition rate for spectral-focusing hyperspectral SRS imaging. The pulse energy on the sample is on the level of pico-Joule. Such low pulse energy together with chirping to picosecond duration effectively avoided sample photodamage, while the high repetition rate allowed fast chemical mapping of molecules adsorbed on gold nanostructured surfaces. Second, we employ a penalized least squares (PLS) approach and successfully extract the sharp Raman peaks from a spectrally broad non-Raman background largely contributed by the photothermal effect[42]. Third, harnessing a block-matching and 4D filtering (BM4D) algorithm to denoise a hyperspectral stack, we are able to generate high-quality single-pixel SRS spectra for statistical analysis of single-molecule

events. By a bianalyte method[43–46], we use two isotopologues of adenine that offer unique vibrational signatures and verify PESRS detection of single molecules with Raman cross-section as low as $10^{-30}$ cm$^2$ per molecule. Furthermore, we demonstrate PESRS imaging of adenine resulting from nucleotide degradation as a stress response of S. aureus cells to starvation.

## Results

**Plasmon-enhanced stimulated Raman scattering spectroscopy.** Adenine adsorbed on gold nanoparticles (Au NPs) aggregation substrates (see Methods) is selected as a proof-of-principle system for the demonstration of PESRS. Adenine is one of the four constituent bases of nucleic acids. The Raman band at 723 cm$^{-1}$ of adenine powder, which has a cross-section of $2.9 \times 10^{-30}$ cm$^2$[47], has been studied for single-molecule detection by surface-enhanced Raman spectroscopy (SERS)[46,47] and surface-enhanced CARS[34]. As shown in Fig. 1a, a pump laser centered at 969 nm and a Stokes laser centered at 1040 nm are employed to induce a PESRS spectrum covering a window ranging from 550 to 850 cm$^{-1}$. Then, 10 µL of a 5 mM aqueous adenine solution is added to ca. 2–4 µL of a concentrated Au colloid suspension, which induces the aggregation of Au NPs. A representative extinction spectrum of an adenine-induced Au NPs aggregation substrate is shown in Fig. 1b. The plasmonic band of the aggregated Au NPs is broad and peaked at 1040 nm, which allows PESRS for the pump and Stokes laser wavelength used here. The resulting PESRS spectrum (Fig. 1c, black) from the adenine-adsorbed Au NPs aggregates consists of a narrower feature at 733 cm$^{-1}$ (highlighted by green) on top of a strong and broad non-resonant background. This sharp feature is close to the prominent adenine ring-breathing mode frequency observed in the normal SRS spectrum of adenine powder (Fig. 1c, blue) and identical to the corresponding 733 cm$^{-1}$ peak observed in the SERS results on Au substrates (Supplementary Fig. 2)[48]. The blank result (Fig. 1c, red) is independently measured from the Au NPs substrate without adenine adsorption. The background could arise from three different non-Raman processes: photothermal effect, cross-phase modulation, and transient absorption[42], all due to laser interactions with the gold nanostructures. The spectral shift between the substrate with/without adenine may relate to the different extent of aggregation with/without adenine. These backgrounds are spectrally overlapped with the SRS signal, but are largely independent of the Raman shift[42]. In contrast, the SRS signal originates from a vibrational resonance that has a sharp spectral feature. A PLS approach is used to fit the broad spectral background. The resulting fitting backgrounds of PESRS are shown in Fig. 1c as the dash lines for the corresponding observed PESRS spectra. Figure 1d shows the vibrationally resonant component of the PESRS spectra resulting from subtraction of the fitting backgrounds from the observed PESRS signals. The PESRS spectrum of adsorbed adenine shows a dominated peak at 733 cm$^{-1}$. Only a noisy baseline is evident after background subtraction from the pure substrate spectrum. Compared with the SRS spectrum of adenine crystal (blue line in Fig. 1d), a 10 cm$^{-1}$ blue shift of the peak is observed in the PESRS spectrum. This blue shifted frequency (733 cm$^{-1}$) is consistent with the strongest vibrational feature observed in SERS spectra of adenine (Supplementary Fig. 2)[49]. These results collectively indicate that the observed vibrational PESRS signal component originates from the surface adsorbed adenine. Figure 1d (purple) presents the standard SRS spectrum of adenine powder at the same laser power condition as used for the detection of PESRS. The standard SRS setup could not generate any Raman signal from a pure adenine powder, while PESRS could detect a thin layer of adenine adsorbed on Au nanostructures. This result indicates that the large electromagnetic field boosted by the plasmon significantly amplified the stimulated Raman process.

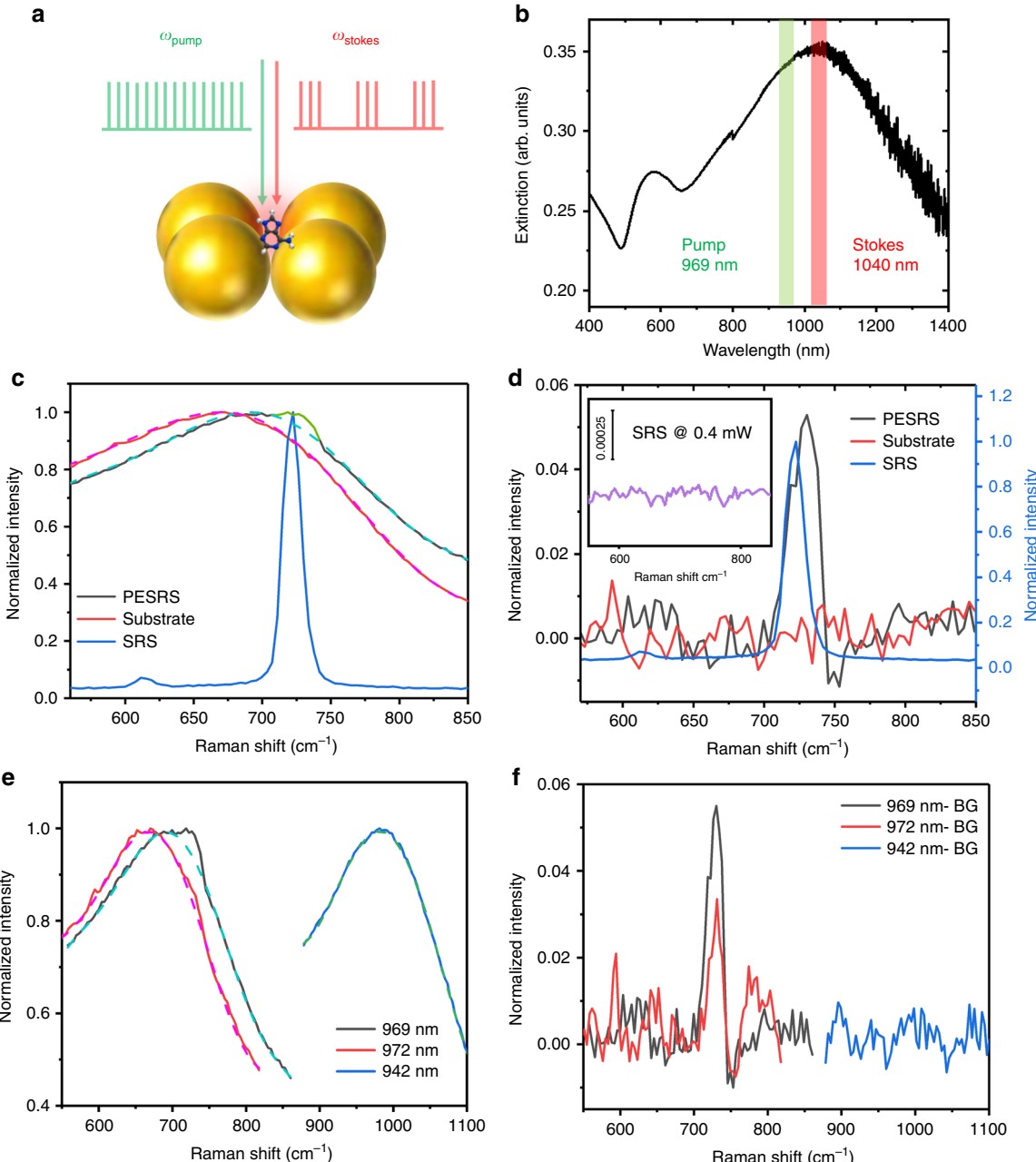

**Fig. 1** PESRS spectroscopy. **a** A schematic of PESRS. **b** A representative extinction spectrum of adenine-induced Au NPs aggregation substrate. **c** The PESRS spectrum (solid) with a green highlighted portion and fitted background (dash) obtained from the substrate with adenine adsorption. The blank spectrum (solid) and fitted background (dash) obtained from the substrate without adenine. The total power of pump and Stokes was 0.4 mW. The SRS spectrum of adenine powder (blue) was obtained with a pump power at 10 mW and a Stoke power at 50 mW. **d** The background-subtracted PESRS spectrum of adsorbed adenine versus the SRS spectrum of adenine powder (same as blue line in **c**) and the spectrum of blank substrate. Inset: The SRS spectrum of adenine powder obtained as the same laser power condition as the PESRS. **e** PESRS spectra (solid) and fitted background (dash) of adenine at Raman resonance (969 nm and 972 nm) and off-resonance (942 nm). **f** Background-subtracted PESRS spectra of adenine at Raman resonance and off-resonance. BG: background.

To verify that the SRS signal is due to the adenine vibrational resonance, we vary the pump wavelength while keeping the Stokes wavelength fixed. The pump laser centered at 972 nm, as well as the previous 969 nm wavelength encompass the adenine Raman resonance for a 1040 nm Stokes pulse, and both generated SRS spectra showing a pronounced peak at the expected wavenumber (Fig. 1e, black and red). In contrast, the 942 nm is off-resonance for the 733 cm$^{-1}$ band. Accordingly, the measured spectrum does not exhibit such a peak as shown in Fig. 1e (blue). After subtraction of the background in Fig. 1e (corresponding

fitted backgrounds were shown as dash lines), the PESRS spectra of adenine excited by both Raman resonance wavelengths show a Raman peak at 733 cm$^{-1}$ (black and red, Fig. 1f), whereas the off-resonance spectrum only shows a noisy featureless baseline (blue, Fig. 1f). Moreover, as shown in Supplementary Fig. 3, the intensity of the 733 cm$^{-1}$ peak linearly depends on the pump power and the Stokes power before it reaches saturation. To evaluate the degree of photodamage, we continually measured a 1-mM adenine PESRS sample at the same location (Supplementary Fig. 4 and Supplementary Movie 1). About 20% of the signal

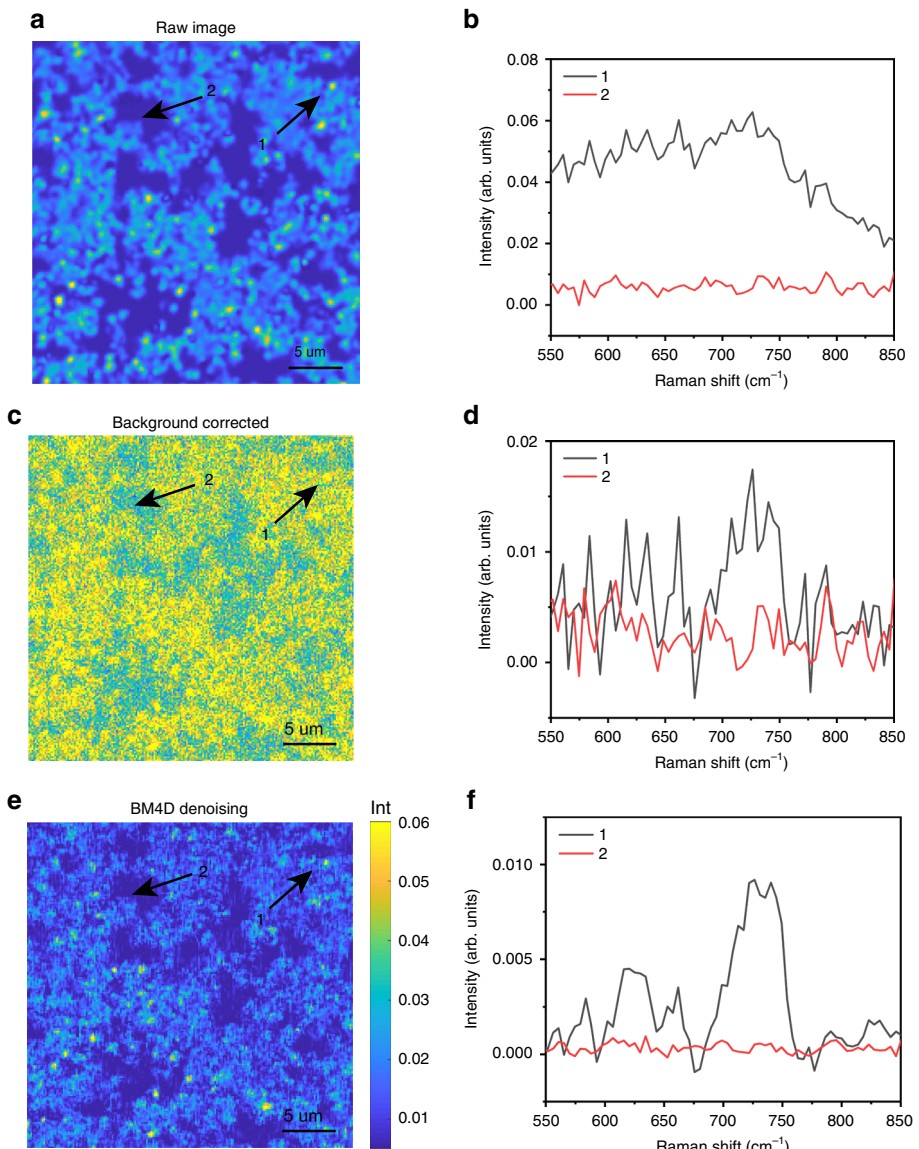

**Fig. 2** Single-pixel PESRS. **a** The raw PESRS image of aggregated Au NPs substrate with adsorbed adenine. The color of each pixel represents the average total spectral channels intensity from each PESRS spectrum. **b** The raw single pixel spectra obtained from spot 1 and 2, which are indicated in **a**. **c** The PESRS image of adsorbed adenine on aggregated Au NPs substrate. The color of each pixel represents the peak area at 733 cm$^{-1}$ after background subtraction. **d** The background-removed single-pixel spectra obtained from spot 1 and 2 which are indicated in **c**. **e** The BM4D-denoised PESRS image of adsorbed adenine on aggregated Au NPs substrate. The color of each pixel represents the peak area at 733 cm$^{-1}$ after BM4D denoising and background subtraction. **f** The BM4D-denoised single-pixel spectra obtained from spot 1 and 2 which are indicated in **e**. Image area: 30 μm × 30 μm.

is lost after 1.5 h continuous exposure. The reproducible spectra recorded at the same location demonstrate that the laser power in our experiment minimally damaged the substrate or induced molecular photodegradation during SRS imaging (~1.0 min per hyperspectral stack). These results collectively confirm the SRS origin of the vibrationally resonant component of the observed spectrum and the plasmonic enhancement of this signal. To ensure that our method is not specific to adenine, we tested other molecules. Supplementary Fig. 5 shows the PESRS spectra of Rhodamine 800 (85 μM in solution) and 4-mercaptopyridine (5.7 mM in solution) adsorbed on the Au NPs aggregated substrate.

**PESRS at single-pixel level**. To demonstrate the imaging capability of PESRS, we scan the adenine containing aggregated Au

NPs substrate with a pixel dwell time of 10 μs. It takes ca. 1 min to obtain a hyperspectral cube (200 × 200 pixel, 80 Raman shifts) consisting of 40,000 spectra. In Fig. 2a, the averaged total 80 spectral channels of an original PESRS hyperspectral data cube are plotted to show the spatial distribution of aggregated NPs. Figure 2b shows two single-pixel spectra from regions with and without NP aggregates, indicated as spot 1 and spot 2, respectively. The single-pixel spectrum from spot 1 shows a broad background and a weak Raman peak around 733 cm$^{-1}$. After pixel by pixel subtraction of the fitting background, the area of the resulting vibrational band at 733 cm$^{-1}$ at each pixel is shown in Fig. 2c revealing a clear spatial contrast between regions of adsorbed adenine and blank areas. The single-pixel background-removed spectra from spot 1 and 2 are displayed in Fig. 2d. It remains challenging to obtain high-quality single-pixel spectra due to the noisy non-Raman background. To address this challenge, we employ a BM4D algorithm which was

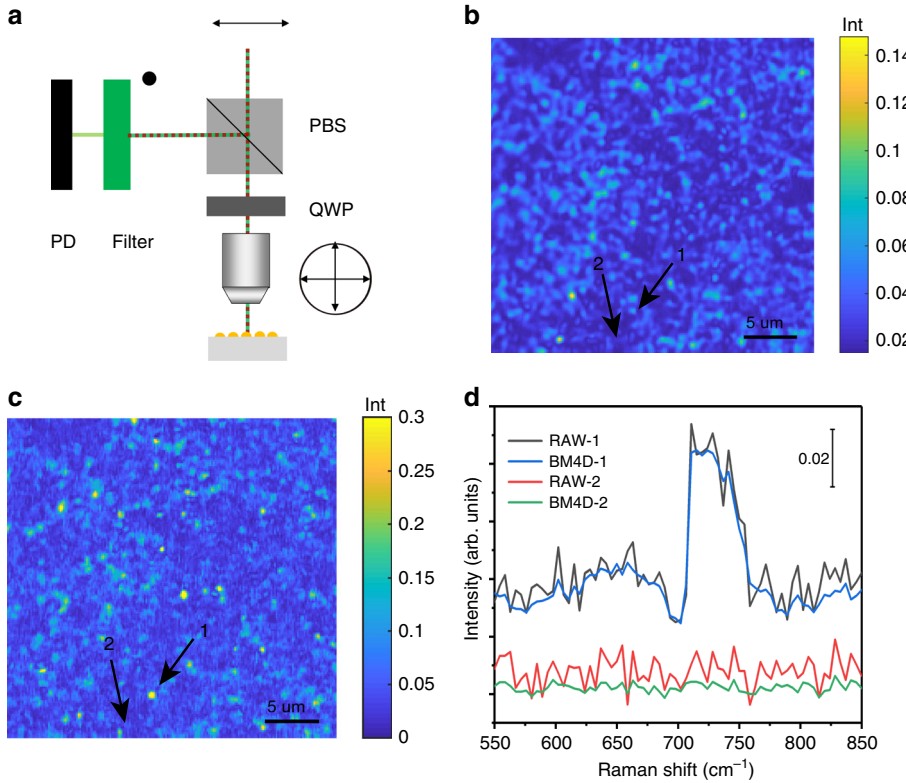

**Fig. 3** Epi-detected PESRS. **a** A schematic of epi-detected PESRS. A polarizing beam splitter (PBS) and a quarter wave plate (QWP) changes the polarization of incoming and backscattered lasers by 90°. In this way, the stimulated Raman loss signal passes the filter and is detected by a photodiode (PD). **b** Raw PESRS image of adenine adsorbed on Au NPs-SiO$_2$ substrate. The color of each pixel represents the average intensity of each PESRS spectrum. **c** Denoised PESRS image of adenine adsorbed on Au NPs-SiO$_2$ substrate. The color of each pixel represents the intensity of the 733 cm$^{-1}$ peak in each denoised and background-corrected PESRS spectrum. The image area is 30 μm × 30 μm. **d** Single-pixel spectra of adenine on the Au NPs-covered SiO$_2$ substrate obtained from spot 1 and 2 indicated in **c**.

widely used for 3D data denoising[50,51]. The reconstructed peak area image and the single-pixel spectra after BM4D denoising and background removal are shown in Fig. 2e, f, respectively. By employing BM4D, we achieve a signal-to-noise ratio of 33 for the single-pixel spectra at spot 1, ~4 times better than that without denoising. Good Raman reproducibility in terms of peak frequency in different locations of the imaging area is found (see Supplementary Fig. 6) even given an expected inhomogeneous intensity distribution due to the Au NPs randomly aggregated hot spots.

PESRS is also demonstrated for epi-detection of molecules on a non-transparent plasmonic substrate that is often used in plasmon-enhanced spectroscopy applications. The experimental setup is shown in Fig. 3a. We use a sol-gel-derived SiO$_2$ substrate covered by immobilized aggregates of monodisperse-sized Au NPs (AuNPs–SiO$_2$ substrate)[52]. Ten microliter of a 100 μM adenine solution are dropped on this plasmonic substrate and dried in air. The spectrally integrated image (Fig. 3b) reveals the distribution of NP clusters on the SiO$_2$ chip. After BM4D denoising and background subtraction, the distribution of hot spots is evident in Fig. 3c. Single-pixel spectra extracted from spots 1 and 2 are indicated in Fig. 3d. After denoising, a signal-to-noise ratio of 48 is achieved for these single-pixel spectra. These data collectively show the high sensitivity of epi-detected PESRS.

To estimate the relative enhancement factor of a local hot spot, we assume a monolayer surface coverage of adenine and a monolayer NP cluster under the laser focus. Based on the measured local PESRS intensity (spot 1) and the average SRS intensity of 5 mM adenine solution, the power-averaged and concentration-averaged local enhancement factor of PESRS relative to normal SRS is estimated to be ~$7 \times 10^7$ (see details

in Supplementary Note 1). Consistent with this result, enhancement factors of $10^4$–$10^6$ and $10^5$–$10^8$ were reported for surface-enhanced femtosecond SRS[39] and surface-enhanced CARS spectroscopy[33,34,36], respectively.

**Single-molecule sensitivity in PESRS.** To quantitate the detection sensitivity of PESRS, we use a well-accepted bianalyte approach developed by the Le Ru[43,44] and Van Duyne[45] groups. The bianalyte approach was developed as a statistically robust method for verification of single molecule detection, as shown in recent single molecule studies in SERS[43–47], and SECARS[34]. The bianalyte approach relies on the observation of two different analytes adsorbed on a nanostructured surface. At the single-molecule level, components of the observed spectra can be attributed exclusively to one or the other of the two analytes by virtue of its distinguishable Raman spectrum. The most straightforward bianalyte approach uses a pair of isotopologues that are chemically identical but spectrally distinct. Here, we use a pair of isotopic molecules of adenine ($^{14}$NA) and adenine-1, 3-$^{15}$N$_2$ ($^{15}$NA). PESRS spectra of ensemble averaged pure $^{14}$NA and pure $^{15}$NA (both 1 mM in solution) show clearly distinguishable Raman bands centered at 733 cm$^{-1}$ and 726 cm$^{-1}$, respectively (Fig. 4a). The PESRS spectrum of an equimolar solution of $^{14}$NA and $^{15}$NA displays two Raman band at 733 cm$^{-1}$ and 726 cm$^{-1}$. The frequency of isotope peaks match well with the SERS measurement (Supplementary Fig. 8) and previous papers[46,47]. These frequency features allow for identification of individual molecules and their mixture in PESRS spectra. Notably, to clearly distinguish the small Raman shift between $^{14}$NA and $^{15}$NA and resolve

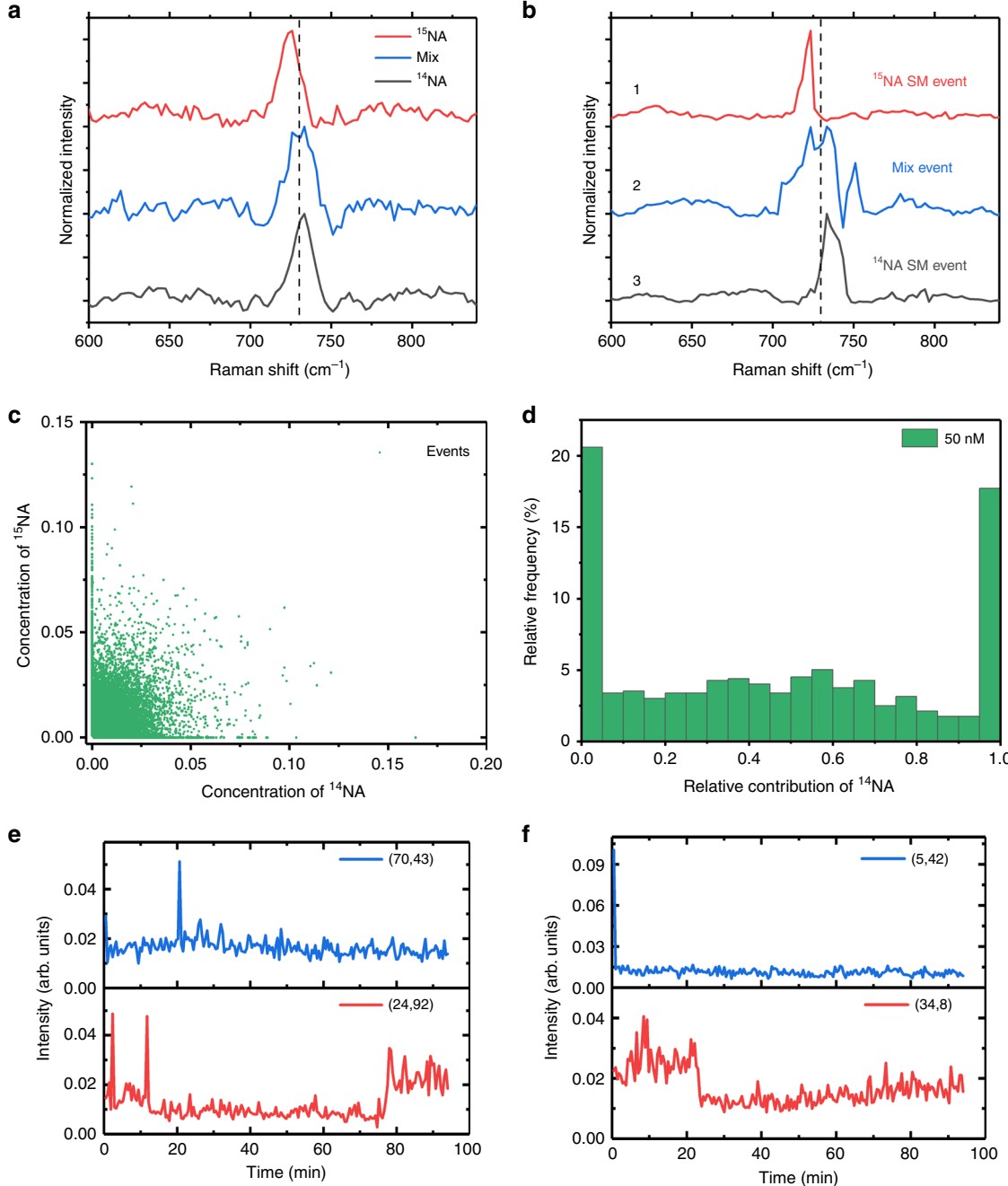

**Fig. 4** Single-molecule sensitivity in PESRS. **a** The ensemble PESRS measurements of pure $^{14}$NA, pure $^{15}$NA and their equimolar mixture. **b** Three representative single-pixel PESRS spectra showing a pure $^{15}$NA SM event (1), a mix event (2), and a pure $^{14}$NA SM event (3), obtained from the 50 nM mixture sample. The vertical dash lines indicate the position of 730 cm$^{-1}$. **c** Concentration matrix coefficients obtained from MCR analysis of the hyperspectral imaging result sample (including 40000 single-pixel spectra) of the mixture. **d** Histogram of relative contribution of $^{14}$NA from the hyperspectral imaging result of the mixture sample. Selected 796 single-pixel spectra with a desired Raman peak and above an intensity threshold were used. SM: single molecule. **e**–**f** Representative time traces of PESRS intensity collected of 50 nM adenine sample showing digital intensity fluctuation (**e**) and single-step photodamage (or photobleach) processes (**f**). The inside labels in **e** and **f** show the X–Y pixel coordinate where the spectra were recorded in Supplementary Movie 2.

the concentration ratio of $^{14}$NA and $^{15}$NA, we create more chirping of the pump and Stokes pulses with two additional rods, which improve the spectral resolution to ca. 7 cm$^{-1}$ (Supplementary Fig. 9) for the single-molecule PESRS study.

To evaluate the sensitivity of PESRS, we prepare a mixture solution of $^{14}$NA and $^{15}$NA at 50 nM concentration each with Au NPs and dry the colloid on a cover glass under vacuum. A hyperspectral PESRS image of this mixture sample, consisting of

40,000 spectra, is acquired for statistical analysis. Figure 4b shows typical single-pixel spectra after denoising and background subtraction (Original corresponding single-pixel spectra can be found in Supplementary Fig. 10). Spectrum 1 has a single peak at 724 cm$^{-1}$, matching the spectrum taken from the reference sample of isotopically pure $^{15}$NA. Spectrum 3 has a single band at 733 cm$^{-1}$, corresponding to the PESRS spectrum of $^{14}$NA. Spectrum 2 exhibits a double band. The left band at 724 cm$^{-1}$

can be attributed to $^{15}$NA, and the right band at 733 cm$^{-1}$ can be attributed to $^{14}$NA. Notably, spectra with single peaks around 726 and 733 cm$^{-1}$ were observed at multiple pixels (see Supplementary Fig. 11). These data allow the statistical analysis described below.

A multivariate curve resolution (MCR) method is used for statistical analysis of the PESRS spectra. The hyperspectral data (containing 40000 single-pixel spectra) is unmixed by MCR into concentration contributions of pure $^{14}$NA ($C_{14}$) and pure $^{15}$NA ($C_{15}$) spectra (Fig. 4c) and a relative concentration ratio of $^{14}$NA $\left(\frac{C_{14}}{C_{14}+C_{15}}\right)$ defined as the fraction of the average number of $^{14}$NA molecules contributing to the total signal, is thus determined. We select 796 of the total spectra acquired that display the desired Raman bands and have an intensity above a threshold value (maximum values > 0.03) for this statistical analysis. The threshold requirement helps to reduce inclusion of noise events and avoid the counting of artificial molecular events[44]. The histogram of relative contributions to the total signal produced by $^{14}$NA is obtained by counting the selected events. As shown in Fig. 4d, the histogram of intensity ratios has the dominant contribution from the single-molecule events at the ratio ≈0 and ≈1. Based on the Le Ru's statistical methodology[44] and our simulation result (Supplementary Note 2), for histograms with distributions like that shown in Fig. 4d, the edges of the histogram (events for the ratio ≈0 or ≈1) represent single-molecule events. The obtained statistical results match well the expected statistics prediction of single-molecule PESRS model data (Supplementary Fig. 12c). As a control experiment, we measure PESRS from 50 nM pure $^{14}$NA and pure $^{15}$NA samples. The histograms of relative contributions of $^{14}$NA of isotopic pure $^{14}$NA and pure $^{15}$NA sample are dominated by events at ratio ≈1 (Supplementary Fig. 13b) and events at ratio ≈0 (Supplementary Fig. 13a), respectively. As a second control, we prepare a high-concentration $^{14}$NA and $^{15}$NA mixture (500 μM each). For this concentrated solution of the two isotopic molecules, mixed events dominated the histogram (Supplementary Fig. 14a). In the pure $^{15}$NA sample (Supplementary Fig. 14b), a significant portion of signals is assigned to pure $^{15}$NA (ratio ≈0) and little to $^{14}$NA. The same result is obtained for the pure $^{14}$NA sample (Supplementary Fig. 14c). We further use the Fano-line shape function to fit the single pixel spectra of the 50 nM adenine sample. Our results (Supplementary Fig. 15) show that the dispersive line shapes of PESRS spectra slightly affect the Raman peak frequency. However, this slight frequency shift has no obvious impact on the molecular assignment between $^{14}$NA and $^{15}$NA.

In addition to spectral segregation, we analyze the bandwidths of single molecule events (the ratio ≈0 or ≈1) versus the mix events (the ratio between 0.3 and 0.7) by fitting the 680–780 cm$^{-1}$ portion of each spectrum to a Fano-line shape fuction[53]. Supplementary Fig. 16 shows the bandwidth distributions of SM $^{14}$NA events and SM $^{15}$NA events are centered at ca. 10 cm$^{-1}$, whereas the bandwidth distribution of mix events is centered at 14 cm$^{-1}$. This bandwidth measurement further distinguishes the single-molecule events from the mixed molecules events.

To further evaluate the single-molecule sensitivity of PESRS, we study the PESRS signal in the temporal domain. Specifically, we record time-lapsed PESRS images from the same area of a 50-nM adenine sample (Supplementary Movie 2). Figure 4e shows digital intensity fluctuations of adenine during the PESRS measurement at the same location. This blinking phenomenon is not observed at 1 mM of adenine (Supplementary Fig. 17). As another evidence, we observe single-step photobleaching at some pixels (Fig. 4f). On the contrary, stable intensity traces are collected from the 1 mM adenine sample (Supplementary Fig. 17).

The spectral blinking and single-step photodamage phenomena are considered as a characteristic behaviors of single or a few molecules[1,2,34,35,54,55]. Collectively these measurements in the spectral and temporal domains support that PESRS allows detection of single-molecule events for biomolecules having a cross-section as low as 10$^{-30}$ cm$^2$.

**PESRS mapping of adenine generated from bacteria**. The investigation of dynamic living samples requires imaging at a high speed. Compared to SERS, the dramatically improved imaging speed of PESRS microscopy makes it a potentially useful tool for imaging the chemical dynamics of a complex living specimen. To demonstrate such capacity, we study the metabolic response of *S. aureus* to starvation, as shown in Fig. 5a. Following enrichment in a nutrient-rich environment, the *S. aureus* sample is washed and centrifuged in pure water. After 1 h, 1 μL of a bacterial suspension is placed on the Au NPs-SiO$_2$ plasmonic substrate and once the water evaporated (~5 min) the PESRS signal is acquired. A control sample is similarly prepared but in contrast a PESRS spectrum is obtained without a 1-h delay. PESRS spectra of *S. aureus* under starvation conditions for 1 h are displayed in Fig. 5b, and the observed spectra closely resemble the Raman spectrum of adenine. In contrast, the spectra of *S. aureus* obtained immediately (no waiting period) (Fig. 5b) do not exhibit an adenine-like Raman band. These results are consistent with the SERS data (Supplementary Fig. 18)[48]. These data imply that adenine, a purine degradation product, is secreted from *S. aureus* as a response to the no-nutrient, water-only environment[48]. As shown previously[48], these molecular species are secreted from the bacterial cells under starvation conditions and appear most heavily concentrated in the pericellular region near the outer cell wall. Figure 5c shows the PESRS image of starved *S. aureus* on the plasmonic substrate, which presents the distribution of the secreted adenine. The two representative single-pixel PESRS spectra of *S. aureus* are presented in Fig. 5d. These results collectively demonstrate that PESRS has the potential for the study of the bacterial exogenous metabolome.

We have noted that SERS is a powerful and easy-to-use method to obtain the single-spot chemical information with high sensitivity. However, point-by-point scanning SERS imaging remains a time-consuming measurement. In previous SERS experiment[48], it took ca. 10 s to obtain a SERS spectrum of bacteria. Thus, for imaging a 30 μm × 30 μm area with 60 × 60 spectra, the total measurement time would be over 600 min, which is much longer than the time scale of metabolic change within the bacteria. Compared with SERS, our PESRS method provides a much faster chemical image. This capacity opens new opportunities for real-time imaging dynamic biological processes, as well as rapid scanning of large areas of tissue labeled by plasmonic Raman tags. Moreover, PESRS imaging can sample millions of pixels in a specimen within a short time, which avoids pixel-dependent fluctuations of signal intensity encountered in SERS spectroscopy and consequently allows quantitative chemical analysis.

## Discussion

Through plasmonic enhancement and hyperspectral recording, the detection sensitivity of SRS microscopy reaches the single-molecule level. The Au plasmonic nanostructures provide an extraordinary SRS intensity enhancement relative to normal SRS of about 10$^7$. Such large enhancements allow the detection of single molecule with a Raman cross-section as low as 10$^{-30}$ cm$^2$. Single-molecule PESRS detection of adenine molecules is verified by an isotope-edited bianalyte method and time-lapsed PESRS

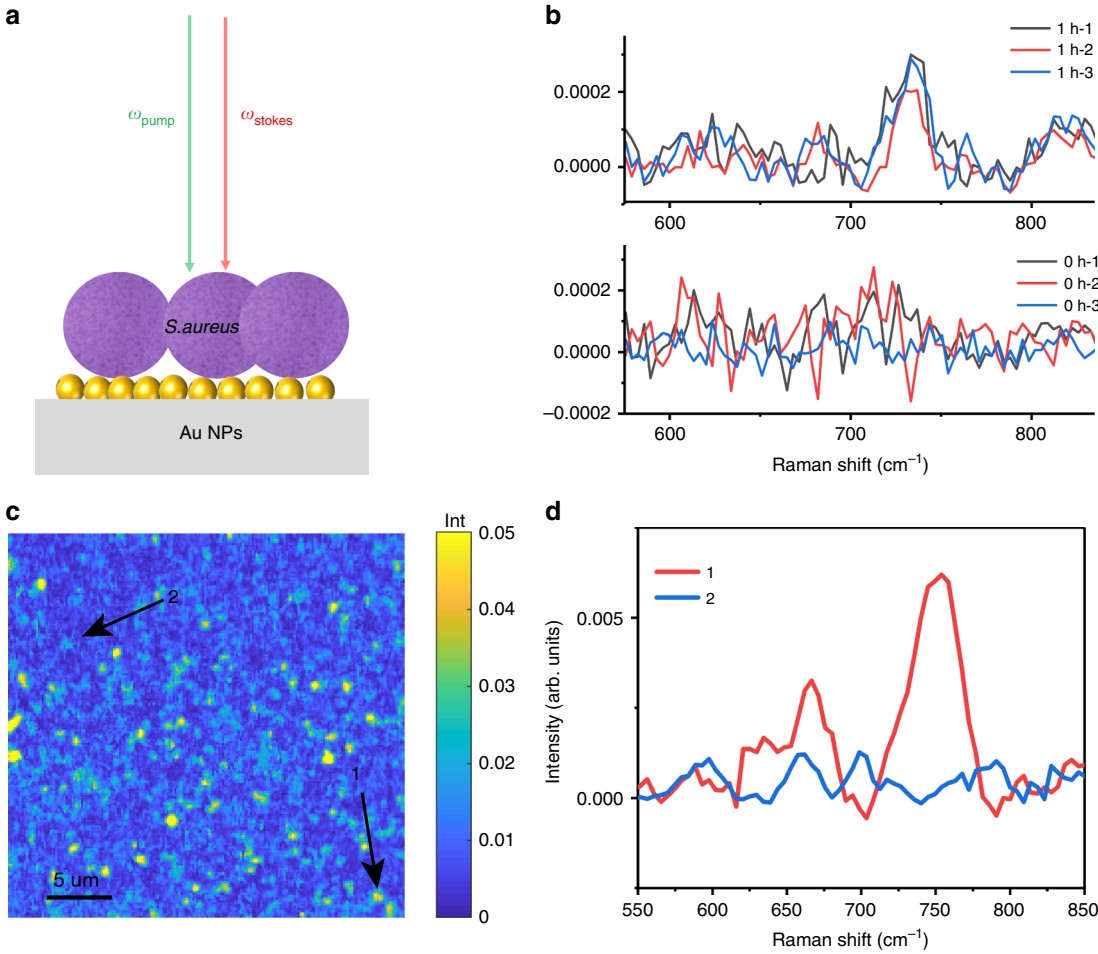

**Fig. 5** PESRS mapping of adenine generated from stressed bacteria. **a** A schematic of PESRS mapping of adenine generated from stressed bacteria. **b** The PESRS spectra of *S. aureus* washed and kept in water for 1 h. The PESRS spectra of *S. aureus* obtained immediately (0 h) after washing. Numbers 1, 2, 3 represent measurements at three locations on surface. No denoising applied. **c** Denoised PESRS image of starved *S. aureus* placed on the plasmonic substrate. The image area is 30 μm × 30 μm. **d** Single-pixel PESRS spectra of *S. aureus* on plasmonic substrate obtained from spot 1 and 2 indicated in panel **c**.

measurements. A potential biomedical application of PESRS is demonstrated through mapping of adenine released from stressed bacterial cells.

In this work, single-molecule detection sensitivity in PESRS is achieved through a combination of several strategies. First, we create and employ a nanostructured Au substrate with a plasmon resonance peak overlapping with our pump and Stokes laser wavelengths. Second, we use chirped pico-Joule laser pulses at 80 MHz repetition rate. Such pulse energy, which is 2 to 3 orders of magnitude lower than that in previous surface-enhanced femtosecond SRS works[39,40], significantly decreased potential photodamage. The high repetition rate also allows high-speed data acquisition. Third, a PLS method is used to distinguish the Raman peak from the broad non-Raman background in the hyperspectral dataset. Finally, a BM4D approach denoises a hyperspectral cube and allows high-quality single-pixel spectra to be obtained for statistical analysis of single molecules events.

A portion of our recorded PESRS spectra shows a dispersive vibrational line shape (Fig. 1f, Supplementary Figs 6 and 11). Such line shape was reported in previous surface-enhanced femtosecond SRS works[39,40,53]. Three possible explanations have been proposed: Fano-resonance of the molecule-nanostructure system[53], interference between the PESRS signal and the plasmon enhanced aggregated NP emission[56], and the effects of the complex character of the plamonic field amplitude[57]. It is

important to note that various dispersive line shapes in single pixel spectra (e.g., Supplementary Figs. 6 and 11) are observed. As previous surface-enhanced femtosecond SRS studies have shown, dispersive line shapes show a strong dependence on the relative position of excitation field with respect to the plasmon resonance[58] and enhancement factor[56]. Our PESRS-active aggregations are composed of randomly clustered nanoparticles. Various line shapes in single pixel spectra illustrate the heterogeneity of localized surface plasmon resonance and local enhancement factor in different PESRS active sites. The distribution and character of PESRS line shapes will be further studied.

PESRS microscopy opens a new window for fast vibrational spectroscopic imaging of low-concentration molecules with high sensitivity. It should be noted that PESRS intensity is strongly dependent on the distance between the molecule and the surface. To achieve the highest sensitivity in PESRS measurements, it is necessary to keep target analytes on the surface with the highest enhancement. In this way, PESRS could sensitively detect the chemical components on a cell membrane or cell wall that is closely attached to the surface. Second, PESRS can detect metabolites secreted from a live cell in the pericellular region and investigate metabolic changes linked to the development of microbial populations or to exposure to antibiotics or other environmental changes. Third, combined with bio-conjugated target-specific plasmonic Raman nanoprobes, hyperspectral

PESRS imaging can be employed for rapid localization of multi-biomarkers in large areas of tissues. In addition, PESRS imaging method can be extended to study the dynamic processes in surface chemistry, such as the mapping of the solid electrolyte interface membrane in a lithium cell or imaging the heterogeneity of catalyst.

Additional improvements may be envisioned for this technology. For example, the imaging speed can be further improved by using a multiplex SRS method[26,27], advanced delay tuning approach[22], or a wide-field SRS system. Secondly, harnessing the rational-designed reproducible plasmonic nanostructure fabricated by lithographic methods, our method can pave the way for reproducible and quantitative molecular imaging platform. Such improvements will invoke the integration of coherent Raman imaging techniques and novel nanostructure designs to open new avenues towards ultra-sensitivity, ultra-fast, and label-free chemical imaging.

## Methods

**Hyperspectral plasmon-enhanced stimulated Raman scattering microscope.** Supplementary Fig. 1 presents the scheme of a hyperspectral SRS microscope. Briefly, an 80 MHz tunable femtosecond laser (InSight DS+, Spectra-Physics) provided the pump (680–1300 nm) and Stokes (1040 nm) stimulated fields. The Stokes beam is modulated by an acousto-optic modulator at 2.3 MHz. The pump and Stokes beams are spatially aligned and sent to an upright microscope with 2D galvo system for laser scanning. The spectral-focusing approach is used to obtain spectral domain information. In spectral focusing, the pump and Stokes pulses are equally stretched in time by 4 glass rods to achieve a constant instantaneous frequency difference that drives a single Raman coherence. By delaying the pump pulses, a series of Raman shifts (80 data points) are generated. At a certain delay, all the laser energy is spectrally focused to excite a narrow Raman band. The laser powers (pump ~0.15 mW and Stokes ~0.15 mW at the sample) are sufficiently low so that good stability of the molecule and nanostructure was maintained during the experiments. A ×60 water immersion objective (Olympus, NA = 1.2) or a ×40 water immersion objective (Olympus, NA = 0.8) is used to focus pump and Stokes laser on a sample. An oil condenser (Nikon, NA = 1.4) is used to collect the laser light in the forward direction. A 1000 nm short-pass filter (Thorlabs) blocks the Stokes laser before a photodiode with a lab-built resonant amplifier. A lock-in amplifier demodulates the stimulated Raman loss signal from the pump beam detected by the photodiode. Using an XY scanner with a 150 nm step to scan the sample, a PESRS hyperspectral data cube (200 × 200 pixels, 80 Raman channels) is recorded with a 10 µs dwell time per pixel. To obtain the PESRS spectrum of adenine, we scan the spectral range from ca. 550 cm$^{-1}$ to 850 cm$^{-1}$ with 13.7 cm$^{-1}$ spectral resolution. To clearly distinguish the small Raman shift between $^{14}$NA, $^{15}$NA and their mixture, two more rods were added in the combined light path to increase the chirp of pump and Stokes. In this way, about 7 cm$^{-1}$ spectral resolution is achieved (as shown in Supplementary Fig. 9) with 120 spectral data points from 565 cm$^{-1}$ to 850 cm$^{-1}$. The PESRS imaging speed is ca. 1 to 2 min per data cube. The image size (30 × 30 µm) and pixel dwell time were the same in all experiments.

An epi-detected SRS microscope is built for PESRS detection on non-transparent plasmon-enhanced substrates. Before the microscope, a quarter wave plate is placed after a polarizing beam splitter to change the polarization of excitation and back-reflected laser light by 90°. In this way, the polarizing beam splitter allows forward light to pass through and the stimulated Raman loss signals are reflected into a photodiode to achieve epi-PESRS imaging (Fig. 3a).

**Background reduction in PESRS.** A raw PESRS spectrum contains a large background signal from the photothermal effect, cross-phase modulation, and transient absorption. Cross-phase modulation originates from the optical Kerr effect, and the transient absorption and photothermal effect are due to the plasmonic resonance of the Au NPs. We minimized the background arising from non-Raman processes by using a larger numerical aperture (NA = 1.4) lens for signal collection to reduce cross-phase modulation and photothermal effect. Moreover, a megahertz-frequency modulation was used to further diminish the photothermal effect. With those approaches, we successfully observe a PESRS signal in the presence of a strong background.

**Background subtraction in a PESRS spectrum.** An adaptive iteratively reweighted penalized least squares (airPLS) algorithm (https://github.com/zmzhang/airPLS), developed by Zhang et al.[59], is employed to subtract the baseline from the raw PESRS spectrum.

**Denoising of a PESRS hyperspectral data.** Firstly, we use a BM4D V3.2 (http://www.cs.tut.fi/~foi/GCF-BM3D/index.html#ref_software) denoising algorithm, developed by Maggioni and Foi[50,51], to process the raw PESRS hyperspectral data cube. The BM4D algorithm relies on the so-called grouping and collaborative

filtering paradigm. A 3D imaging block ($x$–$y$–$\lambda$) is stacked into a 4D data array, which is then filtered. Thus, BM4D leverages the spatial and spectral correlation of a hyperspectral data cube both at the nonlocal and local level. Then, we use the airPLS algorithm to subtract the background of a hyperspectral data cube pixel by pixel. In addition, the image is plotted by the peak area of the desired Raman band.

**Statistical analysis of single-molecule events.** Before the statistical analysis of single-molecule events, the PESRS hyperspectral data is denoised and corrected baseline as described. A multivariate curve resolution (MCR) algorithm, developed by Tauler and Juan et al.[60,61], is used to extract the concentration maps of the two isotopically related molecules in the mixed hyperspectral dataset. Because $^{14}$NA and $^{15}$NA have the same Raman cross-section, we use the normalized spectra separately obtained from pure $^{14}$NA and $^{15}$NA samples as the initial estimation of the pure spectra. The constraints implemented during the optimization step are non-negativity for the concentration and spectrum. The outputs of the MCR treatment are the pure concentration maps ($C_{14}$, $C_{15}$) of $^{14}$NA and $^{15}$NA, respectively. Then, the 796 spectra whose maximum values appeared at the desired wavenumber range are selected and an intensity threshold (maximum values >0.03) is defined for removal of noise events. The histogram of the relative contribution of $^{14}$NA $\left(\frac{C_{14}}{C_{14}+C_{15}}\right)$ is plotted and the edges of the histogram, the ratio ≈0 or ≈1, are considered as the single molecule events. Hyperspectral data unmixing was done by MCR GUI 2.0 (https://mcrals.wordpress.com/download/). All data were processed by MATLAB.

**Substrate preparation.** The Au NPs colloid is prepared according to the classical citrate reduction method[62], resulting in particles with a diameter of ~50 to 60 nm, as shown in Supplementary Fig. 19. The 0.5 mL of 0.01% (g/mL) colloidal suspension is concentrated to 2–4 µL by centrifuging, which is then added to the 10 µL adenine solution. The adenine solution induces the aggregation of Au NPs. The aggregated Au NPs are dropped on a cover glass, followed by vacuum drying to obtain a substrate for PESRS detection.

Au NPs-SiO$_2$ plasmonic substrates are prepared as described previously[52]. Immobilized clustered aggregates of 80 nm Au NPs are grown on a SiO$_2$ chip. A 1 µL adenine solution was dropped on the Au NPs-SiO$_2$ plasmonic substrates and samples are ready for PESRS measurement after adenine solution dried under air ambient (~5 mins). High-purity water (Milli-Q, 18.2 MΩ·cm) is used throughout the study.

**Bacteria sample preparation.** Bacteria are harvested during the log phase. Culture growth media is removed from the bacterial samples by centrifugation, and washing four times with 2 mL of deionized Millipore water. The bacterial pellet is suspended in 0.25 mL of water, and 1 µL of the resulting bacterial suspension is dropped and dried onto the Au NPs-SiO$_2$ plasmonic substrate. Samples are dried onto the Au substrate either immediately or after 1 h in order to demonstrate the effect of the starvation stress response.

Additional experimental results and data analysis are available in the supplementary information.

**Reporting summary.** Further information on research design is available in the Nature Research Reporting Summary linked to this article.

## Data availability
The data that support the plots within this paper and other findings of this study are available from the corresponding author upon reasonable request.

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

## Acknowledgements

This work was supported by NIH grants GM118471, AI141439, and a Keck Foundation grant to J.X.C., Xiamen University postdoctoral fellowship to C.Zo. supported by NSFC (2163000117, 21621091, and 21790354), and MOST (2016YFA0200601) grants to B.R. L.D.Z. acknowledges the support of NSF (CHE-1609952). We thank Jiayingzi Wu for assistance with UV-Vis detection, and Kai-Chih Huang, Fengyuan Deng, and Peng Lin for their valuable suggestion that greatly improved the experimental performance.

## Author contributions

Experiments were designed by J.X.C, C.Zo., and L.D.Z. The PESRS experiments were conducted by C.Zo. Data analysis was executed by C.Zo. with the help of H.N.L. P.R. prepared the Au NPs-SiO₂ substrate and the bacteria sample. Y.M.H. performed the SEM study and UV-Vis detection. H.N.L., C.Zh., and C.Zo. developed the hyperspectral SRS instruments. C.Zo. wrote the paper, revised by J.X.C., C.Y., B.R., and L.D.Z. All authors commented on the paper.

## Competing interests

The authors declare no competing interests.
