## [Peer Review File · Nature Communications]

Reviewers' comments:

Reviewer #1 (Remarks to the Author):

To the best of my knowledge, this manuscript details the first experimental demonstration of plasmon-enhanced stimulated Raman scattering (PESRS) microscopy for high-speed label-free chemical imaging of biomolecules. While the combination of plasmon-enhanced excitation and coherent Raman scattering detection has been widely reported before (e.g.: Refs. [21-26] for plasmon-enhanced CARS, and Refs. [28-30] for plasmon-enhanced SRS), I consider the exploitation of plasmon-enhanced SRS detection sensitivity for the fast multispectral SRS imaging of biomolecules at the surface of a plasmonic substrate without the requirement of electronic pre-resonant excitation as the primary novelty presented in this manuscript. By combining a well-established plasmonic substrate assay with a previously demonstrated concept of fast spectral focusing SRS imaging, this work presents an estimation of an experimental enhancement factor of PESRS, which is consistent with enhancement factors previously obtained for surface-enhanced femtosecond SRS and surface-enhanced CARS spectroscopy. Furthermore, this work successfully demonstrates the application of fast PESRS mapping of adenine generated from bacteria. Regarding the authors' very prominent claim of reaching and achieving single-molecule detection sensitivity using the PESRS concept, unfortunately, a major shortcoming of this manuscript is the lack of the direct proof of detecting an individual molecule. Merely an indirect evidence has been provided based on a statistical, bi-analyte analysis, which only (to use the authors' own words in line 246) "... represents single-molecule events with high probability". Demonstrating a high probability is NOT the same as proving the detection of an individual molecule without unambiguity! Unless well-established single-molecule characteristics for a single-molecule detection event are presented, as for example has been directly observed for an individual molecule using plasmon-enhanced CARS detection in Ref. [24], the authors should revoke any claim of reaching and achieving single-molecule detection sensitivity throughout their manuscript. As is, the manuscript title, abstract, figure captions, and especially their discussion are misleading the reader.

Otherwise, the experimental work was thoroughly performed. The manuscript was, for most parts, clearly written, and is well organized. I recommend publication in Nature Communications only after the authors will have revised their manuscript accordingly, have addressed the deficiencies that I have listed below in more detail, and have provided additional experimental evidence that support their claims. Please understand my comments below as to strengthen any revisions of this manuscript:

- 1) Introduction, lines 44 and 58: Where appropriate, the original seminal literature should be credited!
- 2) Introduction, line 54: Prior to Refs [23, 24], already back in 2005, Koo et al. [Opt. Letts. 30 (2005) 1024, not cited here] reported the very first claim of achieving single-molecule detection sensitivity by means of surface-enhanced CARS. Please make allowance for this early work in your (critical) discussion.
- 3) Introduction, line 55: Regarding the complicated quantification in plasmon-enhanced CARS due to its nonresonant background and distorted line shapes, I think it is not at all justified to present PESRS as a solution to that here: It is clearly evident that similar issues complicate the quantification in PESRS! For example, the interference between the broad nonlinear background and the resonant SRS results in dispersive and broadened lines (cp. Figs. 1d, 1f, and 3d). These observations are even more clearly evident in the single-pixel PESRS spectra shown in Figs. S5 and S17. In addition, the weaker Raman resonances of adenine residing in the range from 800 to 850 cm^{-1} are often not observed at all (also not in your SRS spectrum of adenine powder), while the weak resonance at about 625 cm^{-1} of similar spontaneous Raman intensity is observed! Please provide explanations for these observations and amend your discussion accordingly.
- 4) Epi-detected PESRS, Fig. 3b: Since the raw single-PESRS spectrum from spot 1 in Fig. 3D already indicates a peak intensity that is well above the noise level, the corresponding arrow in Fig. 3b points to a spot where there seems to be no raw pixel intensity! Is that simply a drawing

error?

5) Single-molecule sensitivity in PESRS, line 220 and line 95 in SI: The observation of an isotopologue-specific resonance frequency feature alone does NOT allow for an unequivocal identification of an individual molecule! Also, the spectral variation between different single-pixel PESRS spectra is NOT a signature of single-molecule events! Rather, the temporal fluctuation of spectral features in one and the same single-pixel PESRS spectrum would allow testing a single-molecule detection event. For actually proving the latter, characteristic single-molecule behaviour such as digital changes in spectral features and/or intensity (i.e. blinking) need to be observed. Please provide such additional experimental data and amend your discussion accordingly.

6) Single-molecule sensitivity in PESRS, line 132-135 and Fig. S3: Another commonly accepted signature for detecting an individual molecule is the observation of single-step photodamage. While the latter has been observed for single-molecule events in plasmon-enhanced CARS detection (cp. Ref. [24]), apparently no such observation has been described in this work (even while increasing the pump and Stokes powers). In case photodamage was indeed observed by the authors, what was the observed time profile of PESRS intensity during photodamage? Please provide such additional experimental data and amend your discussion accordingly.

7) Single-molecule sensitivity in PESRS, lines 229-231: To circumvent the limited spectral resolution of their SRS system, which is unfortunately just similar to the difference in isotopologue-specific resonance Raman shifts, a simple increase of chirp in spectral focusing SRS would have helped. Why has that not been implemented?

8) PESRS mapping of adenine generated from bacteria, lines 262-264: The authors emphasize the high-speed PESRS imaging advantage for the investigation of dynamic biological processes, which cannot be obtained by using conventional SERS. To demonstrate and quantify this PESRS advantage, however, a direct experimental comparison for the same sample substrate would be required. Unfortunately, the latter has not been presented by the authors. Furthermore, the presented study of the bacterial exogenous metabolic changes over a time scale of 1 hour rather shows the potential of PESRS mapping in general but does not demonstrate the full advantage in the investigation fast dynamics, which cannot be studied otherwise! At least, please provide a critical discussion of your results that also takes the comparison with the imaging speed in similar SERS studies into account.

9) The estimation of local enhancement factor of PESRS: If the enhancement factor (EF) of PESRS relative to normal SRS is defined by $EF = I_{\text{PESRS}}/I_{\text{SRS}}$ and by using the definition of intensities in line 68 (in the SI), then the equation in line 73 (in the SI) seems to be erroneous. Please double-check.

Reviewer #2 (Remarks to the Author):

In this paper, the authors demonstrated plasmon-enhanced SRS microscopy with a single molecule sensitivity. By using analytes as a linker for metal nanoparticle aggregation, the strong enhancement of Raman scattering has been achieved, which realized single-molecule SRS detection. The use of spectral focusing for spectrum detection and spectrum processing using PLS and BM4D successfully extracted the SRS spectrum from the background given by the photothermal and other effects without vibrational resonance. The paper is well written with a quality high enough to be published in Nature Communications. However, I would like to request the following two things to validate the authors' results and novelty further.

1. Data without background subtraction and denoising in single molecule detection

From Fig.1 and S4, it is clear that the authors' approach worked well for the high-concentration samples. Since single molecule detection gives a lower signal, and it would be fair to show the same data set (spectrum with and without background subtraction and denoising). This is helpful for readers to see the robustness of the measurement.

2. The necessity of spectrum detection

It seems that one of the keys for PESRS is post-processing, which requires spectrum detection.

From this point, It is important to mention the spectrum range and data points required to perform PESRS.

2. Comparison with spontaneous Raman

Compared with spontaneous Raman with plasmon resonance, PESRS requires much more effort to show the spectra. In addition, the single-band SRS does not work with PE due to the necessity of spectrum processing. From those points, the benefit of SRS seems not much appreciate in plasmon enhanced approach. It would be useful if the authors could give comparisons with the use of spontaneous Raman.

Reviewer #3 (Remarks to the Author):

This article by Zong et al, a collaborative effort by experts in the plasmonics (Ziegler) and SRS fields (Cheng), reports on single molecule sensitivity in plasmon-enhanced simulated Raman spectroscopy, as evidenced by the isotopologue proof approach. The work is also supplemented by application of PE-SRS imaging to adenine detection from stressed bacteria, although not at the single molecule level.

Single molecule sensitivity in SRS is a significant advance, as SRS is free from the background issues which plague CARS, the only other coherent Raman technique in which plasmon-enhanced single molecule sensitivity has been claimed. To that end, the results in this work rest upon the significance of the histogram in Figure 4D. At first glance, this is a remarkably convincing isotopologue proof. However, I have serious concerns about the data analysis, in particular the background subtraction, fitting, and frequency resolution, which call into question these claims. Unless these issues can be sufficiently resolved, I would not recommend the manuscript for publication.

1. First, this histogram is noticeably better than any other ever observed for single molecule SERS. One could argue that SRS is more sensitive, but given the overall low signal magnitudes and the relatively high concentrations of analytes used for deposition here, it is quite surprising to expect such a dramatic result. This requires further explanation.

2. It is strange that the authors used the normal Raman frequencies for assignment of the isotope peaks, rather than a high concentration single isotope SERS spectrum. The peaks shift frequency significantly in the presence of gold, and this impacts the accuracy of the fitting.

3. The spectral resolution of these measurements, and more importantly how it affects the fitting, is not described. This is particularly important when looking at the spectra in the SI, which are quite noisy. There has to be some significant error associated with assigning a particular peak to ^{14}N or ^{15}N , which does not seem to be accounted for. Given the noise level in the "representative" spectra, I don't see how the histogram in figure 4D could be obtained without significant errors in the fitting.

4. Providing a similar histogram for the peak width as a function of ^{15}N concentration would help, as presumably the peaks would be wider for mixed events as compared to single molecule events.

5. The lack of consistent lineshapes in the spectra is quite concerning, and it seems likely that dispersive lineshapes are not correctly accounted for in the algorithm. For example, many of the spectra in Figure S5 and S7 look dispersive. How do the fits account for lineshape or Fano q parameter? This could significantly affect the assigned frequency, which could impact claims of single molecule sensitivity. The authors also do not provide enough explanation for this phenomena in the text, as all previous SE-SRS measurements have shown dispersive lineshapes, which vary depending on the plasmon resonance frequency. The plasmon resonance frequency

effect is also not taken into account here, which would require correlated LSPR measurements.

6. The inclusion of the bacterial imaging is a bit strange, as it does not really relate to single molecule SRS detection. The spectra are also extremely noisy.

7. A major limitation to plasmon-enhanced spectroscopy is the rapid decay of enhancement with distance from the surface on the 1-10 nm length scale. The authors need to discuss this limitation in the context of biological PE-SRS imaging.

Response letter to reviewers

Reviewers' comments:

Reviewer #1 (Remarks to the Author):

To the best of my knowledge, this manuscript details the first experimental demonstration of plasmon-enhanced stimulated Raman scattering (PESRS) microscopy for high-speed label-free chemical imaging of biomolecules. While the combination of plasmon-enhanced excitation and coherent Raman scattering detection has been widely reported before (e.g.: Refs. [21-26] for plasmon-enhanced CARS, and Refs. [28-30] for plasmon-enhanced SRS), I consider the exploitation of plasmon-enhanced SRS detection sensitivity for the fast multispectral SRS imaging of biomolecules at the surface of a plasmonic substrate without the requirement of electronic pre-resonant excitation as the primary novelty presented in this manuscript. By combining a well-established plasmonic substrate assay with a previously demonstrated concept of fast spectral focusing SRS imaging, this work presents an estimation of an experimental enhancement factor of PESRS, which is consistent with enhancement factors previously obtained for surface-enhanced femtosecond SRS and surface-enhanced CARS spectroscopy. Furthermore, this work successfully demonstrates the application of fast PESRS mapping of adenine generated from bacteria.

Regarding the authors' very prominent claim of reaching and achieving single-molecule detection sensitivity using the PESRS concept, unfortunately, a major shortcoming of this manuscript is the lack of the direct proof of detecting an individual molecule. Merely an indirect evidence has been provided based on a statistical, bi-analyte analysis, which only (to use the authors' own words in line 246) "... represents single-molecule events with high probability". Demonstrating a high probability is NOT the same as proving the detection of an individual molecule without unambiguity! Unless well-established single-molecule characteristics for a single-molecule detection event are presented, as for example has been directly observed for an individual molecule using plasmon-enhanced CARS detection in Ref. [24], the authors should revoke any claim of reaching and achieving single-molecule detection sensitivity throughout their manuscript.

As is, the manuscript title, abstract, figure captions, and especially their discussion are misleading the reader. Otherwise, the experimental work was thoroughly performed. The manuscript was, for most parts, clearly written, and is well organized. I recommend publication in Nature Communications only after the authors will have revised their manuscript accordingly, have addressed the deficiencies that I have listed below in more detail, and have provided additional experimental evidence that support their claims. Please understand my comments below as to strengthen any revisions of this manuscript:

1) Introduction, lines 44 and 58: Where appropriate, the original seminal literature should be credited!

A1 (Answer 1): We appreciate the reviewer's suggestion. We added the following original references (Ref 12-15) to the revised manuscript.

Ref.12 Cheng, J.-X. et al. *J. Phys. Chem. B* **108**, 827 (2004).

Ref 13 Evans, C.L. et al. *Proc. Natl. Acad. Sci. U. S. A.* **102**, 16807-16812 (2005)

Ref 14 Freudiger, C.W. et al. *Science* **322**, 1857-1861 (2008).

Ref 15 Saar, B.G. et al. *Science* **330**, 1368-1370 (2010).

2) Introduction, line 54: Prior to Refs [23, 24], already back in 2005, Koo et al. [Opt. Letts. 30 (2005) 1024, not cited here] reported the very first claim of achieving single-molecule detection sensitivity by means of surface-enhanced CARS. Please make allowance for this early work in your (critical) discussion.

A2: We appreciate the reviewer's suggestion. We cited Koo's paper (Opt. Letts. 30, 2005, 1024) properly as Ref 26.

3) Introduction, line 55: Regarding the complicated quantification in plasmon-enhanced CARS due to its nonresonant background and distorted line shapes, I think it is not at all justified to present PESRS as a solution to that here: It is clearly evident that similar issues complicate the quantification in PESRS! For example, the interference between the broad nonlinear background and the resonant SRS results in dispersive and broadened lines (cp. Figs. 1d, 1f, and 3d). These observations are even more clearly evident in the single-pixel PESRS spectra shown in Figs. S5 and S17.

A3-1: We appreciate the reviewer's comment. Indeed, the dispersive shape of SRS could complicate the quantification ability of PESRS. We delete the related statement.

In addition, the weaker Raman resonances of adenine residing in the range from 800 to 850 cm^{-1} are often not observed at all (also not in your SRS spectrum of adenine powder), while the weak resonance at about 625 cm^{-1} of similar spontaneous Raman intensity is observed! Please provide explanations for these observations and amend your discussion accordingly.

A3-2: We appreciate the reviewer's comment. Based on our experimental data (as shown in Figure S9b) and previous papers (e.g. N. J. Halas, *J. Phys. Chem. C* 2009, 113, 14390, and Kim, M. S., *J. Raman Spectrosc*, 1986, 17(5), 381.), there is not a clear spontaneous Raman peak in the range from 800 to 850 cm^{-1} . Similar results are shown in the SERS spectrum of adenine (as shown in Fig S2) and the previous paper (W. R. Premasiri et al, *Anal Bioanal Chem* (2016) 408:4631–4647). This is the reason that why we cannot observe the peak in the range from 800 to 850 cm^{-1} in our SRS and PESRS spectra.

Figure S9. (b) The measured spontaneous Raman spectrum of adenine powder

4) Epi-detected PESRS, Fig. 3b: Since the raw single-PESRS spectrum from spot 1 in Fig. 3D already indicates a peak intensity that is well above the noise level, the corresponding arrow in Fig. 3b points to a spot where there seems to be no raw pixel intensity! Is that simply a drawing error?

A4: We appreciate the reviewer's comment. We redrew the corresponding arrow in Fig 3b, as shown below.

Figure 3. Epi-detected PESRS. (a) Schematic. A polarizing beam splitter (PBS) and a quarter wave plate (QWP) changes the polarization of incoming and backscattered lasers by 90°. In this way, the stimulated Raman loss signal passes the filter and is detected by a photodiode (PD). (b) Raw PESRS image of adenine adsorbed on Au NPs-SiO₂ substrate. The color of each pixel represents the average intensity of each PESRS spectrum. (c) Denoised PESRS image of adenine adsorbed on Au NPs-SiO₂ substrate. The color of each pixel represents the intensity of the 733 cm⁻¹ peak in each denoised and background-corrected PESRS spectrum. The image area is 30 μm × 30 μm. (d) Single-pixel spectra of adenine on the Au NPs-covered SiO₂ substrate obtained from spot 1 and 2 indicated in (c).

5) Single-molecule sensitivity in PESRS, line 220 and line 95 in SI: The observation of an isotopologue-specific resonance frequency feature alone does NOT allow for an unequivocal identification of an individual molecule! Also, the spectral variation between different single-pixel PESRS spectra is NOT a signature of single-molecule events! Rather, the temporal fluctuation of spectral features in one and the same single-pixel PESRS spectrum would allow testing a single-molecule detection event. For actually proving the latter, characteristic single-molecule behaviour such as digital changes in spectral features and/or intensity (i.e. blinking) need to be observed. Please provide such additional experimental data and amend your discussion accordingly.

A5: We appreciate the reviewer's comment. Many single molecule verification experiment, such as spectral blinking, single-step photodamage, Poisson distribution of intensity, polarization properties, and bianalyte approach, have been developed. Here, we used a well-accepted method, isotope-edited bianalyte, to explore the single-molecule sensitivity. In the SERS community, the bianalyte and isotopologue approach was developed as a statistically robust method for single-molecule detection, as shown in many recent single molecule studies in SERS (Ref 37-41) and SECARS (Ref 28).

However, to follow referee's suggestions, we further validated the single molecule sensitivity of PESRS by monitoring the temporal fluctuation of spectral features. We measured the time-dependent PESRS signals (Movie S2) from a 50 nM adenine solution adsorbed on Au NPs as shown in Figure S15-1. In contrast to more concentrated solutions of adenine which always yield a stable intensity trace, we observed nearly digital Raman intensity fluctuation in the 50 nM adenine sample. This spectral blinking phenomenon is considered an additional signature of the behavior of single molecules spectral sensitivity.

Figure S15-1 Representative time traces of PESRS spectra collected of 50 nM adenine solution (a) showing digital intensity fluctuation and 1.0 mM adenine (b) showing relative stable intensity traces. The inside labels show the pixel positions where the spectra were recorded.

6) Single-molecule sensitivity in PESRS, line 132-135 and Fig. S3: Another commonly accepted signature for detecting an individual molecule is the observation of single-step photodamage. While the latter has been observed for single-molecule events in plasmon-enhanced CARS detection (cp. Ref. [24]), apparently no such observation has been described in this work (even while increasing the pump and Stokes powers). In case photodamage was indeed observed by the authors, what was the observed time profile of PESRS intensity during photodamage? Please provide such additional experimental data and amend your discussion accordingly.

A6: We appreciate the reviewer's comment. In the temporal measurement of PESRS, we indeed observed single-step photodamage of PESRS signal in 50 nM adenine adsorbed on Au NPs as shown in Figure S15-2. As a control experiment, the ensemble sample (Figure S15-1b) shows a stable signal.

50 nM adenine

Figure S15-2 Representative time traces of PESRS spectra collected on (a) 50 nM adenine which shows single-step photodamage (or photobleach) processes.

7) Single-molecule sensitivity in PESRS, lines 229-231: To circumvent the limited spectral resolution of their SRS system, which is unfortunately just similar to the difference in isotopologue-specific resonance Raman shifts, a simple increase of chirp in spectral focusing SRS would have helped. Why has that not been implemented?

A7: We appreciate the reviewer's suggestion. We added 2 more rods in the combined path to increase the degree of pulse chirp in our spectral focusing SRS for single molecule PESRS measurement. As shown in Figure S9 below, spectral resolution is significantly improved to be $\sim 7 \text{ cm}^{-1}$, c.a. 2 times better than what we presented in the first submission, and is comparable to a commercial Raman microscope (Figure S9b). At the improved spectral resolution, we measured a series of concentration ratio of $^{15}\text{NA}/^{14}\text{NA}$ (Figure S9c&d). Our result indicates that the spectral resolution (7 cm^{-1}) of our PESRS microscope has the ability to distinguish the different concentration ratios of ^{15}NA to ^{14}NA . The single molecule results presented in the revised manuscript were measured with the high spectral resolution setting.

Figure S9. Spectral resolution of spectral focusing SRS microscopy. (a) The SRS spectra and corresponding FWHM of adenine powder measured by different chirped laser. 3+1 rods: 3 rods in combined light path and 1 rod in Stokes light path. 5+1 rods: 5 rods in combined light path and 1 rod in Stokes light path. (b) The spontaneous Raman spectrum of adenine powder measured by Renishaw inVia Raman microscope with 1200/mm grating. Laser: 633 nm. (c) PESRS spectra as a function of concentration ratio of ^{15}NA and ^{14}NA . (d) PESRS peak positions of mixture samples as a function of concentration ratio of ^{15}NA and ^{14}NA . With the increasing of ^{14}NA , the PESRS peaks of mixture shift to high wavenumber. The PESRS spectra were measured by 5+1 rods. This result indicates that with a spectral resolution of 7 cm^{-1} , our PESRS microscope is able to distinguish the concentration ratio of ^{15}NA and ^{14}NA .

8) PESRS mapping of adenine generated from bacteria, lines 262-264: The authors emphasize the high-speed PESRS imaging advantage for the investigation of dynamic biological processes, which cannot be obtained by using conventional SERS. To demonstrate and quantify this PESRS advantage, however, a direct experimental comparison for the same sample substrate would be required. Unfortunately, the latter has not been presented by the authors. Furthermore, the presented study of the bacterial exogenous metabolic changes over a time scale of 1 hour rather shows the potential of PESRS mapping in general but does not demonstrate the full advantage in the investigation fast dynamics, which cannot be studied otherwise! At least, please provide a critical discussion of your results that also takes the comparison with the imaging speed in similar SERS studies into account.

A8: We appreciate the reviewer's comment. In a previous SERS study of bacteria (W. R. Premasiri et al., *Anal Bioanal Chem* (2016) 408, 4631), it took 10 s to obtain a SERS spectrum of

starved bacteria. For SERS imaging of 30×30 μm with 0.5 μm scan step, it will take at least 600 min to obtain a 60×60 pixel SERS image. The SERS imaging time is much longer than the time scale of bacterial metabolic change. With PESRS imaging, we only took 1 min to obtain a 200×200 pixel chemical imaging with 150 nm scan step. The PESRS imaging can provide a much faster and much finer chemical image than SERS. To emphasize the advantage of fast imaging of PESRS, we revised the discussion as following:

“We have noted that SERS is a powerful and easy-to-use method to obtain the single-spot chemical information with high sensitivity. However, point-by-point scanning SERS imaging remains a time-consuming measurement. In previous SERS experiments, it took c.a. 10 s to obtain a SERS spectrum of bacteria. Thus, for imaging a 30 μm × 30 μm area with 60 × 60 spectra, the total measurement time would be over 600 min, which is much longer than the time scale of metabolic change within the bacteria. Compared with SERS, our PESRS method provides a much faster chemical image. This capacity opens new opportunities for real-time imaging dynamic biological processes as well as rapid scan of larger area of tissue labeled by plasmonic Raman tags. Moreover, PESRS imaging can sample millions of pixels in a specimen within a short time, which avoids pixel-dependent fluctuations of signal intensity encountered in SERS spectroscopy and consequently allows quantitative chemical analysis.”

9) The estimation of local enhancement factor of PESRS: If the enhancement factor (EF) of PESRS relative to normal SRS is defined by $EF=I_PESRS/I_SRS$ and by using the definition of intensities in line 68 (in the SI), then the equation in line 73 (in the SI) seems be erroneous. Please double-check.

A9: We appreciate the reviewer’s comment. In our work, the enhancement factor (EF) of PESRS relative to normal SRS is defined as a ratio of PESRS over normal SRS cross-sections ($EF = \sigma_{PESRS}/\sigma_{SRS}$). We can use the power- and concentration- averaged intensity between PESRS and SRS to calculate the EF. To avoid misunderstanding, we modified the paragraph (SI7) as following:

“In stimulated Raman scattering, the signal intensity is calculated as: $I = N \times \sigma \times P \times S$, where I is the intensity, N is the number of molecules under the laser spot, σ is the molecular Raman scattering cross-section, P is the pump laser power, and S is the Stokes laser power. Enhancement factor (EF) of PESRS relative to normal SRS is defined as a ratio of PESRS over SRS cross-sections ($\sigma_{PESRS}/\sigma_{SRS}$). To estimate the EF of PESRS, we calculated the power- and concentration-averaged intensity between PESRS and SRS as following:

$$\sigma_{PESRS} = \frac{I_{PESRS}}{N_{PESRS} \times P_{PESRS} \times S_{PESRS}}$$

$$\sigma_{SRS} = \frac{I_{SRS}}{N_{SRS} \times P_{SRS} \times S_{SRS}}$$

Thus ,

$$EF = \frac{\sigma_{PESRS}}{\sigma_{SRS}} = \frac{I_{PESRS}}{I_{SRS}} \times \frac{N_{SRS}}{N_{PESRS}} \times \frac{P_{SRS}}{P_{PESRS}} \times \frac{S_{SRS}}{S_{PESRS}}$$

Reviewer #2 (Remarks to the Author):

In this paper, the authors demonstrated plasmon-enhanced SRS microscopy with a single molecule sensitivity. By using analytes as a linker for metal nanoparticle aggregation, the strong enhancement of Raman scattering has been achieved, which realized single-molecule SRS detection. The use of spectral focusing for spectrum detection and spectrum processing using PLS and BM4D successfully extracted the SRS spectrum from the background given by the photothermal and other effects without vibrational resonance. The paper is well written with a quality high enough to be published in Nature Communications. However, I would like to request the following two things to validate the authors' results and novelty further.

1. Data without background subtraction and denoising in single molecule detection
From Fig.1 and S4, it is clear that the authors approach worked well for the high-concentration samples. Since single molecule detection gives a lower signal, and it would be fair to show the same data set (spectrum with and without background subtraction and denoising). This is helpful for readers to see the robustness of the measurement.

A1: We appreciate the reviewer's comment. We added the original spectra without background subtraction, their fitting background and denoising of single molecule results in SI10, as shown in Figure S10:

Figure S10. The corresponding single-pixel PESRS spectra of ^{15}NA single molecule event (a), Mix event (b), and ^{14}NA single molecule event (c) without denoising, after denoising and fitting background in Figure 4b

2. The necessity of spectrum detection. It seems that one of the keys for PESRS is post-processing, which requires spectrum detection. From this point, It is important to mention the spectrum range and data points required to perform PESRS.

A2: We appreciate the reviewer's comment. The spectrum range is from 550 to 850 cm^{-1} which contains 80 data points. In single molecule measurement, we increased the chirp of pump and Stokes laser for a high spectral resolution. The spectrum range is from 565 to 850 cm^{-1} which contains 120 data points. In the Methods section, we emphasize the spectrum range and data points in PESRS as following:

“Using an XY scanner with a step size of 150 nm to scan the sample, a PESRS hyperspectral data cube (200×200 pixels, 80 Raman channels) was recorded with 10 μ s dwell time per pixel. To obtain the PESRS spectrum of adenine, we scanned the spectral range from c.a. 550 cm^{-1} to 850 cm^{-1} with 13.7 cm^{-1} spectral resolution. To distinguish the small Raman shift between ^{14}NA , ^{15}NA and their mixture, two more rods were added to the light path to increase the chirp of both pump and Stokes beams. In this way, 7 cm^{-1} spectral resolution was achieved (as shown in **SI9**), with 120 spectral data points from 565 cm^{-1} to 850 cm^{-1} .”

3. Comparison with spontaneous Raman. Compared with spontaneous Raman with plasmon resonance, PESRS requires much more effort to show the spectra. In addition, the single-band SRS does not work with PE due to the necessity of spectrum processing. From those points, the benefit of SRS seems not much appreciate in plasmon enhanced approach. It would be useful if the authors could give comparisons with the use of spontaneous Raman.

A3: We appreciate the reviewer’s comment. SERS indeed is a powerful and easy-to-use method to obtain the single-spot chemical information with a high sensitivity. However, as a point-scanning measurement, SERS imaging is a time-consuming process. In previous SERS experiment (W. R. Premasiri et al., *Anal Bioanal Chem* (2016) 408, 4631), it took c.a. 10 s to obtain a SERS spectrum of bacteria. For imaging a $30 \mu\text{m} \times 30 \mu\text{m}$ area with 60×60 spectra, the total measurement time would be over 600 min, which is much longer than the time scale of metabolic change of bacteria. Compared with SERS, PESRS provides a much faster chemical image. It opens new opportunities for real-time imaging dynamic biological processes and for rapid scanning larger area of tissue labeled by plasmonic Raman tags. Moreover, PESRS imaging can sample millions of pixels in a specimen within a short time. This capacity avoid pixel-dependent fluctuation of signal intensity encountered in SERS spectroscopy and consequently allows quantitative chemical analysis. To emphasize the advantage of fast PESRS imaging, we modified the paragraph and made a comparison between SERS and PESRS on page 16.

Reviewer #3 (Remarks to the Author):

This article by Zong et al, a collaborative effort by experts in the plasmonics (Ziegler) and SRS fields (Cheng), reports on single molecule sensitivity in plasmon-enhanced simulated Raman spectroscopy, as evidenced by the isotopologue proof approach. The work is also supplemented by application of PE-SRS imaging to adenine detection from stressed bacteria, although not at the single molecule level.

Single molecule sensitivity in SRS is a significant advance, as SRS is free from the background issues which plague CARS, the only other coherent Raman technique in which plasmon-enhanced single molecule sensitivity has been claimed. To that end, the results in this work rest upon the significance of the histogram in Figure 4D. At first glance, this is a remarkably convincing isotopologue proof. However, I have serious concerns about the data analysis, in particular the background subtraction, fitting, and frequency resolution, which call into question

these claims. Unless these issues can be sufficiently resolved, I would not recommend the manuscript for publication.

1. First, this histogram is noticeably better than any other ever observed for single molecule SERS. One could argue that SRS is more sensitive, but given the overall low signal magnitudes and the relatively high concentrations of analytes used for deposition here, it is quite surprising to expect such a dramatic result. This requires further explanation.

A1: We appreciate the reviewer's comment. We have now studied 50 nM mixture samples of ^{14}NA and ^{15}NA . We note that a similar concentration (100 nM) was used in previous single molecule SECARS work (Ref 28). We have obtained the histogram of the relative contribution of ^{14}NA as a convincing bianalyte proof (as shown in Figure 4d).

2. It is strange that the authors used the normal Raman frequencies for assignment of the isotope peaks, rather than a high concentration single isotope SERS spectrum. The peaks shift frequency significantly in the presence of gold, and this impacts the accuracy of the fitting.

A2: We appreciate the reviewer's comment. We indeed used high concentration single isotope PESRS peaks (Figure 4a) to assign the isotope peaks. The PESRS peaks (733 cm^{-1} for ^{14}NA and 726 cm^{-1} for ^{15}NA) match well the previous SERS studies (Ref40 and Ref41) and the high concentration single isotope SERS peaks (735 cm^{-1} for ^{14}NA and 727 cm^{-1} for ^{15}NA) as shown in SI8.

3. The spectral resolution of these measurements, and more importantly how it affects the fitting, is not described. This is particularly important when looking at the spectra in the SI, which are quite noisy. There has to be some significant error associated with assigning a particular peak to ^{14}NA or ^{15}NA , which does not seem to be accounted for. Given the noise level in the "representative" spectra, I don't see how the histogram in figure 4D could be obtained without significant errors in the fitting.

A3: We appreciate the reviewer's comment. To clearly distinguish the small Raman shift between ^{14}NA and ^{15}NA , we improved the spectral resolution of our SRS system by a factor of two via increasing the chirp of the incident lasers pulse. As shown below, the spectral resolution for single molecule detection is about 7 cm^{-1} which is comparable to a commercial Raman microscope. Furthermore, $\sim 2\text{ cm}^{-1}$ frequency precision can be detected, which is much smaller than the Raman shift difference between ^{14}NA and ^{15}NA (c.a. 8 cm^{-1}). We measured a series of the mixture of ^{14}NA and ^{15}NA at different ratios as shown below. Our PESRS results indicated that at the spectral resolution of 7 cm^{-1} , our PESRS microscope has the ability to distinguish the concentration ratios of ^{15}NA and ^{14}NA .

Figure S9. Spectral resolution of spectral focusing SRS microscopy. (a) The SRS spectra and corresponding FWHM of adenine powder measured by different chirped laser. 3+1 rods: 3 rods in combined light path and 1 rods in Stokes light path. 5+1 rods: 5 rods in combined light path and 1 rods in Stokes light path. (b) The spontaneous Raman spectrum of adenine powder measured by Renishaw inVia Raman microscope with 1200/mm grating. Laser: 633 nm. (c) PESRS spectra as a function of concentration ratio of ^{15}NA and ^{14}NA . (d) PESRS peak positions of mixture samples as a function of concentration ratio of ^{15}NA and ^{14}NA . With the increasing of ^{14}NA , the PESRS peaks of mixture shift to high wavenumber. The PESRS spectra were measured by 5+1 rods. This result indicated that with a spectral resolution of 7 cm^{-1} , our PESRS microscope is able to distinguish the concentration ratio of ^{15}NA and ^{14}NA .

4. Providing a similar histogram for the peak width as a function of ^{15}NA concentration would help, as presumably the peaks would be wider for mixed events as compared to single molecule events.

A4: We appreciate the reviewer's comment. As shown below, we statistically analyzed the bandwidth of single molecule spectra and mix events spectra. Figure S14 shows the distribution of peak bandwidth of SM ^{14}NA events, SM ^{15}NA events are centered on c.a. 10 cm^{-1} , and bandwidth of mix events are centered on c.a. 14 cm^{-1} . The bandwidth of mixed events (blue) are larger than the bandwidth of single molecule spectra (black and red).

Figure S14. Bandwidth analysis. (a) The representative single-pixel PESRS spectra (dash lines) of ^{15}NA single molecule event, mix event, and ^{14}NA single molecule event fitted with a Fano-lineshape function (red lines). The fitting function was shown as following:

$$f(x) = A \left\{ \frac{\left(q + \frac{x-x_0}{\Gamma/2} \right)^2}{1 + \left(\frac{x-x_0}{\Gamma/2} \right)^2} \right\}. \text{ Here, } A \text{ is the amplitude of each peak, } q \text{ is the Fano asymmetry}$$

parameter, x_0 is the center frequency of vibrational feature, Γ is the line width. Corresponding fitted q value indicated in text. (b) Histogram displaying the width of the peak bandwidth for SM ^{14}NA (black) events, SM ^{15}NA (red) events, and mix (blue) event. The bandwidth of mixed events (blue, average: $14.6 \pm 6.5 \text{ cm}^{-1}$) is larger than the bandwidths of single molecule spectra (black and red, average: $10.5 \pm 3.9 \text{ cm}^{-1}$).

5. The lack of consistent lineshapes in the spectra is quite concerning, and it seems likely that dispersive lineshapes are not correctly accounted for in the algorithm. For example, many of the spectra in Figure S5 and S7 look dispersive. How do the fits account for lineshape or Fano q parameter? This could significantly affect the assigned frequency, which could impact claims of single molecule sensitivity. The authors also do not provide enough explanation for this phenomena in the text, as all previous SE-SRS measurements have shown dispersive lineshapes, which vary depending on the plasmon resonance frequency. The plasmon resonance frequency effect is also not taken into account here, which would require correlated LSPR measurements.

A5: We appreciate the reviewer's comment. As previous SE-FSRS studies have shown, dispersive line shapes show a strong dependence on the relative position of excitation field with respect to the plasmon resonance (Van Duyne's JPCL 2017, 8, 3328.) and enhancement factor (LD Ziegler, JPCC, 2016, 120,20998). Our PESRS active aggregations were composed of multiple randomly nanoparticles clusters. Various line shapes in different single pixel spectra illustrated the heterogeneity of local LSPR and local enhancement in different PESRS active sites, which would require further correlated local LSPR measurements. We added a detail explanation in the related paragraph in Discussion section (page 19).

To verify the reliability of MCR method to analyze Fano-shape spectra, we measured 1 mM pure ^{14}NA , ^{15}NA and their mixture sample and generated similar histograms of relative concentration of ^{14}NA by MCR method as shown in Fig S13-1. As we expected, the histogram of

500 μM of mixture sample (Fig S13-1a) looks like a Gaussian distribution centered at the ratio=0.5. The histogram of pure ^{14}NA sample (Fig S13-2c) was dominated by pure ^{14}NA signal (ratio ≈ 1). While the “mixture” signal was observed in the histogram. Part of the pure molecule signals were assigned as mixture signals. This assignment may result from the various dispersive line shape of PESRS in different single pixel spectra. In the pure ^{15}NA sample result (Fig S13-2b), most of signals were assigned as the pure ^{15}NA (ratio ≈ 0).

Figure S13-1. The histograms of the relative contribution of ^{14}NA in 500 μM the mixture of ^{14}NA and ^{15}NA sample (a), 1 mM ^{15}NA sample (b) and 1 mM ^{14}NA sample (c).

To further verify the reliability of MCR method to analyze single molecules, we measured 50 nM of pure ^{14}NA and pure ^{15}NA samples, respectively. Fig S12-2a&b shows the histograms of two isotopic pure samples. The histogram of pure ^{14}NA and pure ^{15}NA sample dominated by the ratio ≈ 1 (Fig S12-2b) and ≈ 0 (Fig S12-2a), respectively.

Figure S12-2. The histograms of the relative contribution of ^{14}NA in (a) the 50 nM ^{15}NA sample and (b) 50 nM ^{14}NA sample.

In addition, we used the Fano-line shape function $f(x) = A \left\{ \frac{(q + \frac{x-x_0}{\Gamma/2})^2}{1 + (\frac{x-x_0}{\Gamma/2})^2} \right\}$ to fit the 50 nM adenine spectra. We find that most of fitted Fano asymmetry parameter (q) are larger than 2 or smaller than -2 (Fig S13-2a). Based on this result, we simulated a series of q -dependent ^{14}NA and

¹⁵NA spectra as shown in Fig S13-2b. The simulation result shows that for $q > 2$ or < -2 , the different q values slightly shift the peak position, while ¹⁴NA and ¹⁵NA spectra can be differentiated. These results indicate that the Fano resonance did not have an obvious impact on the molecular assignment.

Figure S13-2. (a) The distribution of q value of PESRS spectra of 50 nM adenine. (b) Simulation ¹⁴NA and ¹⁵NA spectra with a function of Fano q parameter. Peak position $x_0=733 \text{ cm}^{-1}$ and 726 cm^{-1} for ¹⁴NA and ¹⁵NA, respectively. Width $\Gamma=10 \text{ cm}^{-1}$.

6. The inclusion of the bacterial imaging is a bit strange, as it does not really relate to single molecule SRS detection. The spectra are also extremely noisy.

A6: We appreciate the reviewer's comment. In a previous SERS study of bacteria (W. R. Premasiri et al., *Anal Bioanal Chem* (2016) 408, 4631), it took 10 s to obtain a SERS spectrum of starved bacteria. For SERS imaging of $30 \times 30 \mu\text{m}$ with $0.5 \mu\text{m}$ scan step, it will take at least 600 min to obtain a 60×60 pixel SERS image. The SERS imaging time is much longer than the time scale of bacterial metabolic change. With PESRS imaging, it took 1 min to obtain a 200×200 pixel chemical imaging with 150 nm scan step. We include this data to show that PESRS imaging can provide a much faster chemical image than SERS imaging. We have further discussed potential applications of PESRS imaging in the Discussion section of the revised manuscript (page 19).

7. A major limitation to plasmon-enhanced spectroscopy is the rapid decay of enhancement with distance from the surface on the 1-10 nm length scale. The authors need to discuss this limitation in the context of biological PE-SRS imaging.

A7: We appreciate the reviewer's comment. PESRS intensity is strongly dependent on the distance between the molecule and the surface. To achieve the highest sensitivity in PESRS measurements, it is necessary to keep target analytes on the surface with the highest enhancement. In this way, PESRS could sensitively detect the chemical component on a cell membrane or cell wall that is closely attached to the surface. Second, PESRS can detect metabolites secreted from a live cell in the pericellular region and investigate metabolic changes

linked to the development of microbial populations or to exposure to antibiotics or other environmental changes. Third, combined with bio-conjugated target-specific plasmonic Raman nanoprobe, hyperspectral PESRS imaging can be employed for rapid localization of multi-biomarkers in large areas of tissues. In the revised manuscript, we discussed the potential impact of PESRS imaging in the Discussion section (page 19).

Reviewers' comments:

Reviewer #1 (Remarks to the Author):

In their revised manuscript, for most parts, the authors have addressed my questions and concerns adequately. Regarding the original references now added to the manuscript, seminal literature on SRS microscopy that should also be credited is still missing. Otherwise, by taking my suggestions into account, the repetition of experiments at increased spectral resolution and the lower concentration of 50 nM, and the new data of monitoring of PESRS-signals as a function of time significantly improved the manuscript. Based on these additional experiments, the authors' most prominent claim of reaching and achieving single-molecule detection sensitivity using their PESRS concept can now be accepted.

Though, in view of the importance of their new results obtained from the analysis of the time-lapsed PESRS images collected on a 50 nM adenine sample, which the authors currently only hide in the supplementary Figures S15-1 and S15-2, these new data should be presented more prominently in a combined figure in the main text, too! In fact, the authors already fully discuss these data in their main text, and admit themselves now by stating in line 281: "... Collectively these measurements in the spectral and temporal domains support that PESRS allows detection of single-molecule events..." I fully agree with this use of the key word "collectively"! (In fact, supporting evidence for my previously expressed concerns about the residual unambiguity of the bianalyte method to "proof" single-molecule detection has now been presented by the authors themselves in form of the additional control experiments on pure samples shown in Figure S12-2.)

In summary, I recommend publication in Nature Communications only after the authors will have addressed the above-mentioned minor deficiencies and revised the presentation in their manuscript accordingly.

Reviewer #2 (Remarks to the Author):

The authors addressed all the comments and requests from the reviewer and clarified them in the manuscript. I recommend publication of this manuscript in Nature Communications.

Reviewer #3 (Remarks to the Author):

This revised manuscript by Zong et al addresses many of the previous comments by all three reviewers. In particular, the increased spectral resolution achieved by spectral focusing is appreciated, as well as the discussion on the denoising algorithm. Achieving single molecule sensitivity with SRS is an important contribution to the scientific community, and is worthy of publication in Nature Communications.

However, as also mentioned by reviewer 1, the work presented here is not absolute proof of single molecule sensitivity, it only shows that there is a high likelihood of single molecule sensitivity. Thus the title and abstract are highly misleading. I would suggest the authors modify the title to say "with near single-molecule sensitivity", or "with likely single-molecule", in order to accurately convey the results of the paper without overstating the conclusions. Given the maturity of the SERS field, careful and cautious statements are quite important in not mis-leading non-experts.

Response letter for Reviewers' comments

Reviewer #1 (Remarks to the Author):

In their revised manuscript, for most parts, the authors have addressed my questions and concerns adequately. Regarding the original references now added to the manuscript, seminal literature on SRS microscopy that should also be credited is still missing. Otherwise, by taking my suggestions into account, the repetition of experiments at increased spectral resolution and the lower concentration of 50 nM, and the new data of monitoring of PESRS-signals as a function of time significantly improved the manuscript. Based on these additional experiments, the authors' most prominent claim of reaching and achieving single-molecule detection sensitivity using their PESRS concept can now be accepted.

A1: We appreciate the reviewer's suggestion. We modified the manuscript as following and added the following seminal literatures on SRS microscopy to the revised manuscript.

"Hyperspectral SRS microscopy has been achieved by many strategies, such as wavelength tuning (Ref19,20), spectral-focusing (Ref21,22), optical frequencies coding (Ref 23) et al, which provide spectral profile at each image pixel and enable the discoveries of new biology (Ref 24,25)."

Ref.19: Ozeki, Y. *et. al.* Nat. Photonics 2012, 6, 845.

Ref.20: Wang, P. *et. al.* Angew. Chem. Int. Ed. 2013, 52, 13042.

Ref.21: Fu, D. *et. al.* J. Phys. Chem. B 2013, 117, 4634.

Ref.22: Liao, C.-S. *et. al.* Optica 2016, 3, 1377-1380.

Ref.23: Liao, C.-S. *et. al.* Science advances 2015, 1, e1500738.

Ref. 24: Li, J.; *et. al.* Cell Stem Cell 2017, 20, 303-314. e305.

Ref.25: Fu, D. *et. al.* Nature Chem. 2014, 6, 614

Though, in view of the importance of their new results obtained from the analysis of the time-lapsed PESRS images collected on a 50 nM adenine sample, which the authors currently only hide in the supplementary Figures S15-1 and S15-2, these new data should be presented more prominently in a combined figure in the main text, too! In fact, the authors already fully discuss these data in their main text, and admit themselves now by stating in line 281: "... Collectively these measurements in the spectral and temporal domains support that PESRS allows detection of single-molecule events..." I fully agree with this use of the key word "collectively"! (In fact, supporting evidence for my previously expressed concerns about the residual unambiguity of the bianalyte method to "proof" single-molecule detection has now been presented by the

authors themselves in form of the additional control experiments on pure samples shown in Figure S12-2.)

A2: We appreciate the reviewer's suggestion. We modified the Figure 4 and presented the time-lapsed PESRS images collected on a 50 nM adenine which confirmed the single molecule sensitivity of PESRS in the temporal domain.

In summary, I recommend publication in Nature Communications only after the authors will have addressed the above-mentioned minor deficiencies and revised the presentation in their manuscript accordingly.

A3: We appreciate the reviewer's recommendation.

Reviewer #2 (Remarks to the Author):

The authors addressed all the comments and requests from the reviewer and clarified them in the manuscript. I recommend publication of this manuscript in Nature Communications.

A: We appreciate the reviewer's recommendation.

Reviewer #3 (Remarks to the Author):

This revised manuscript by Zong et al addresses many of the previous comments by all three reviewers. In particular, the increased spectral resolution achieved by spectral focusing is appreciated, as well as the discussion on the denoising algorithm. Achieving single molecule sensitivity with SRS is an important contribution to the scientific community, and is worthy of publication in Nature Communications.

However, as also mentioned by reviewer 1, the work presented here is not absolute proof of single molecule sensitivity, it only shows that there is a high likelihood of single molecule sensitivity. Thus the title and abstract are highly misleading. I would suggest the authors modify the title to say "with near single-molecule sensitivity", or "with likely single-molecule", in order to accurately convey the results of the paper without overstating the conclusions. Given the maturity of the SERS field, careful and cautious statements are quite important in not misleading non-experts.

A: We appreciate the reviewer's recommendation.